# A qualitative analysis of the COVID-19 vaccination rollout in Lagos, Nigeria: Client and provider perspectives on the plan, the process and the progress

Oluchi Kanma-Okafor[1]*, Yetunde Odusolu[2], Akin Abayomi[3], Faisal Shuaib[4], Moji Adeyeye[5], Ibrahim Mustapha[6], Segun Ogboye[3], Dayo Lajide[3], Hussein Abdur-Razzaq[3], Ukamaka Okafor[7], Uchenna Elemuwa[5], Akin Osibogun[1,2,6]

1 Department of Community Health and Primary Care, College of Medicine, University of Lagos, Lagos, Nigeria, 2 Department of Community Health and Primary Care, Lagos University Teaching Hospital, Lagos, Nigeria, 3 Lagos State Ministry of Health, Lagos, Nigeria, 4 National Primary Health Care Development Agency, Abuja, Nigeria, 5 National Agency for Food and Drug Administration and Control, Abuja, Nigeria, 6 Lagos State Primary Health Care Board, Lagos, Nigeria, 7 Pharmacists Council of Nigeria, Lagos Office, Lagos, Nigeria

☯ These authors contributed equally to this work.
‡ These authors also contributed equally to this work
* okanma-okafor@unilag.edu.ng

**Data Availability Statement:** All the data gathered for this research is available on request but cannot

## Abstract

Mass vaccination has proven useful in the control of COVID-19, though vaccine rollout has met major challenges. The learning curve of this process has been valuable. This qualitative study aimed to assess the plan, the process and the progress of the COVID-19 vaccination rollout in Lagos, Nigeria. This study was conducted at vaccination centers in eight of the 20 Local Government Areas in Lagos State from May to July 2021 among healthcare administrators, health workers and vaccine recipients. Data were collected by conducting three key informant interviews, 24 in-depth interviews and eight focus group discussions to explore the vaccination experiences of participants and the challenges facing the vaccination plan and process. The interviews and discussions were recorded, transcribed verbatim and analyzed using the thematic approach. The four-phased plan for the vaccine rollout was clear to all the key informants because the vaccination process was preceded by training. The process was strengthened by the electronic registration system, though riddled by the frequently unstable electronic and internet data capturing. This was mitigated by a stopgap manual registration and recording of client details. Challenges in the logistics of maintaining supplies of the disposable materials required for the vaccination process were overcome by the creativity of the health professionals. Vaccine hesitancy, fueled by misinformation, myths and misconceptions about the vaccine and its side effects, played a huge role in the community response. The reported vaccine side effects were mild; fever, headaches, pain at the injection site, excessive eating and sleepiness. Though the COVID-19 vaccination process appeared to have largely made progress, the future of vaccination in Nigeria is predicated upon a bottom-up approach to programmatic planning, health education and local vaccine production. Collaborations such as public-private partnerships have the

be made publicly available for ethical reasons, to protect the participants' privacy. The public deposition would breach compliance with the protocol approved by our research ethics board and would compromise participants' privacy as they would easily be identified. The name of the ethics committee or Institutional Review Board that provided ethical approval for our study is Health Research Ethics Committee, Lagos University Teaching Hospital, Idiaraba, Lagos, Nigeria. A non-author point of contact that is able to receive queries regarding data access is Prof. Oyinkansola Sofola (+234 8033031483, osofola@unilag.edu.ng.

**Funding:** Funding for this study was provided by the Lagos State Government. The funders had no role in the study design, data collection and analysis, decision to publish, or preparation of the manuscript.

**Competing interests:** The authors have declared that no competing interests exist.

potential of boosting vaccine provision for Nigeria's large population to ensure equitable access to vaccines.

## Introduction

The coronavirus pandemic since its onset in 2019 has seen efforts the world over towards its control, including the production of vaccines. Successful vaccine rollout would interrupt the natural spread of the virus and lead to an end to the COVID-19 pandemic. The global effort toward COVID-19 vaccination has been monumental, shattering the past records of the fastest vaccine project of four and a half years (1963–1967) for the mumps vaccine [1]. The first mass vaccination programme started in the UK in early December 2020 and since then, the number of doses of the vaccine available globally for administration has gradually improved. At the time of writing, the number of doses of the COVID-19 vaccines secured globally was given as 17.5 billion [2]. The careful process of facilitating the vaccination response against COVID-19 included carefully examining the data available from many parts of the world about the vaccine. This was necessitated by the reports that the use of the AstraZeneca vaccine had been interrupted in many countries, mostly in Europe, as a precautionary measure because some recipients of the vaccine were said to have experienced blood clots as a side effect [3]. However, these reports about the formation of blood clots in recipients of the vaccine have since been proven to be false [4,5]. Vaccination against COVID-19 has been slow in Africa especially for the countries relying on doses from the COVAX initiative, a partnership between The Coalition for Epidemic Preparedness Innovations (CEPI), Gavi the Vaccine Alliance, The United Nations Children's Fund (UNICEF) and The World Health Organisation (WHO) and from India where the local demand is being prioritized. Of the 1.3 billion COVID-19 vaccine doses administered globally, Africa currently has received only 1% [6]. Africa faces the overwhelming threat of being left behind as the proportions of the global COVID-19 vaccine doses received in Africa seem to be reducing [6].

While some countries in Africa like Botswana, Ethiopia and Ghana have had exemplary COVID-19 vaccine rollouts, some countries have administered less than half of the doses they have received [7]. Delays in the complete rollout and administration of vaccines have been attributed to factors such as inadequately trained personnel, a lack of funds to support the vaccination programme, poor planning and vaccine hesitancy based on myths and misinformation [6], although the apparently slow rollouts may be due to incorrect reporting.

In early March 2021, Nigeria received close to 4 million doses of the AstraZeneca/Oxford vaccine, COVID-19 vaccine manufactured by the Serum Institute of India (SII), through the COVAX facility, the vaccine coordinating arm of the Access to COVID-19 Tools (ACT) Accelerator [8]. The ACT Accelerator is an innovative collaboration working globally to ensure the speedy development and production of COVID-19 tests, treatments, and vaccines and their equitable accessibility to countries all over the world [9]. To enable the commencement of the COVID-19 vaccination in Nigeria, the vaccines were subsequently distributed to all the 36 States of the federation and the Federal Capital Territory [10]. To aid vaccine rollout IT systems were deployed, including vaccine-tracking systems (such as CDC's VTrckS) and uniquely designed immunization information systems (IIS), necessary for allocating, distributing, recording, and monitoring the vaccines as they are distributed to various parts of the state. These posed an additional challenge in terms of operation and cybersecurity bearing in mind the sensitive nature of the patient data, to ensure privacy is protected. Some COVID-19-vaccine developers and regulators have already fallen victim to cyberattacks [11,12].

The Coronavirus vaccination campaign in Lagos State, Nigeria began on the 12th of March 2021 after receiving over 500,000 doses of the AstraZeneca vaccine, ten days after the vaccines provided for use in Nigeria were available for distribution to the States [13]. Vaccines were distributed to the Local Government Areas. Vaccine distribution was stratified according to the population of each Local Government Area. The vaccination process was based on a four-phase plan that recognised priority groups and the use of the T.E.A.C.H strategy for the COVID-19 vaccination plan [14,15]. T.E.A.C.H is an acronym for a five-point strategy used for the implementation of vaccination, where T stands for Traditional method of vaccinating target populations using desk review of available data sources, identifying the vaccination sites and rolling out, E stands for Electronic self-registration for health workers and the public using an internet link which provides an online form, A stands for Assisted electronic registration, C stands for Concomitant e-registration for walk-in patients at fixed sites or health facilities, H stands for House-to-house registration using volunteers as an additional push to rapidly increase the e-registration [16].

Health workers and front-line responders began receiving the vaccine days after it arrived in Lagos on the 12th of March, 2021 [13], as in many other countries, where the rollout prioritizes essential workers [17]. Whilst Phase 1 of the vaccine rollout plan was initially conceived to include only healthcare workers, other front-line workers, ports of entry (air, land, and seaports), Military, COVID-19 rapid response team (RRT), laboratory network, policemen, petrol pump workers and strategic leaders, it was reviewed to include teachers, members of the judiciary and adults over 70 years of age. Gradually any adult above the age of 18 years was eligible for vaccination [18].

## Aim

This study provides a realistic risk assessment of vaccine deployment to inform future COVID-19 vaccination success by qualitatively assessing the plan, process and progress of the vaccine rollout in Lagos Nigeria.

## Materials and methods

This study was conducted from May to July 2021.

### Background of the study area

Lagos State is situated in the South-Western part of Nigeria, It is divided into three senatorial districts made up of five administrative divisions namely; Ikeja, Badagry, Ikorodu, Lagos and Epe. It has 20 Local Government Areas (LGA) (16 urban and 4 rural) and 37 Local Council Development Areas (LCDA). The population of Lagos State in 2021 was estimated at 14,862,111 [19]. There are 4 public tertiary healthcare facilities, 31 secondary healthcare facilities and 277 Primary Healthcare Centres (PHC) in Lagos State. Of the PHCs, 57 are equipped as priority facilities known as 'flagship' centres to provide 24 hours comprehensive care services. These flagship facilities have referral linkages to the higher levels of care. Vaccine rollout in Lagos State was organised at the primary level of care. However, the COVID-19 vaccination in Lagos State was conducted in 88 health facilities across the state, which included PHCs and General Hospitals [20].

### Study sites

The study was conducted at selected health facilities designated as COVID-19 vaccination centres and the administrative offices where the key informants in the vaccine plan were located.

## Study design

This was a descriptive cross-sectional qualitative study.

## Study population

The study was conducted among clients (vaccine recipients) at the designated COVID-19 vaccination centres, the frontline workers and administrators in the COVID-19 vaccination process in Lagos State.

## Sampling methodology

**Site selection.**   This was conducted by multistage sampling from the list of 88 COVID-19 vaccination sites in Lagos [21]. The LGAs where the COVID-19 vaccine was available were first stratified into urban and rural LGAs. Next, 7 LGAs from the urban LGAs and 1 LGA from the rural LGAs were selected by ballot, a total of 8 LGAs. Also, by ballot one vaccination site from each of the 8 selected LGAs was included in the study. Eight (8) sites in all were selected for the study.

**Selection of discussants, interviewees and key informants.**   The study participants were purposively selected based on their involvement in the vaccine rollout programme. The clients were recruited at the vaccination centres right after they had received either the first or second dose of the vaccine after the study was explained to them and they gave consent. The interviewees and key informants were formally approached, by prior request via a letter, to be interviewed.

## Data collection tools and techniques

Data collection was entirely a qualitative inquiry, from the clients through focus group discussions (FGDs), and from the frontline workers and key informants through in-depth interviews (IDIs) and key informant interviews (KIIs) respectively. Data about the experiences of vaccine recipients during the vaccination process and their understanding of the vaccine rollout phases collected through FGDs were moderated using an open-ended, easy-to-understand, non-judgmental, FGD guide. An unstructured moderator's guide was used to explore providers' perspectives on the vaccine rollout plan, process and progress through the IDIs and KIIs. The FGD guide and the moderator's guide were produced from the knowledge of already existing problems related to the COVID -19 vaccine roll-out and from previous studies [22,23].

An FGD was preferred over individual interviews for the vaccine recipients so as to gather information beyond clients' knowledge, attitudes and practice related to the COVID-19 vaccine and also explore their shared attributes and differences, if any, which otherwise would be impossible to examine [24]. The specific frontline workers required for our study in each PHC do not number enough to make up an FGD.

## Focus group discussion

Eight (8) FGDs, with 8 clients in each group, were carried out, one in each of the selected PHCs. Men and women were included in each FGD. The discussions were held soon after the clients received the vaccine, on the same day of vaccination, in adequately spaced rooms or open spaces that provide privacy, comfort, good ventilation, and lighting. The facilitator guided the discussion to ensure that the discussion was engaging with each person participating adequately. Before the discussion, each participant provided written informed consent and was assigned a unique identification number to protect their privacy. The proceedings of each focus group discussion were recorded using an audio recorder. A note-taker documented the

non-verbal expressions, moods and other physical gestures. Each FGD lasted for 30–45 minutes.

### In-depth interviews

Three (3) IDIs were conducted in each of the selected Local Government Areas among the 3 main front-line health workers in PHC. In all, 24 IDIs were conducted. Each IDI lasted for about 35–50 minutes.

### Key informant interviews

Three (3) KIIs were conducted among the 3 key officers in the Lagos State Ministry of Health. Each KII lasted for about 25 minutes.

### Training of research assistants

The investigators (KOJ and OY) and research assistants underwent training on conducting FGDs, KIIs and IDIs. The training emphasized correct data handling and maintaining respondents' confidentiality,

### Data analysis

The FGC, KII and IDI recordings were transcribed verbatim in entirety capturing the nuances of observations and participants' descriptions. Data transcripts were developed immediately after data were collected. The transcripts were reviewed by a team of four of the authors (OJK, YO, HA, and AO). Thematic analysis was done based on the framework approach [25]. Two of the authors present at data collection and two that were not present ensured transferability, considering the cultural or professional context of the participants [26]. The authors read the transcripts several times. First, the verbatim transcripts were read and reread for overall understanding, and so that the authors familiarized themselves with the details of the transcripts. The transcripts were then examined and coded to search for patterns (content analysis) identifying relationships and similar phrases based on a theoretical framework. The theoretical framework summarized the key concepts and theories surrounding vaccines, the rollout process, vaccine uptake and the factors mitigating the success of the exercise describing any possible relationships between these concepts. The framework is based on a literature review [6,7,10,16]. Next, by the process of indexing [27], the authors extracted identified units and interpretations from the transcripts. Thematic analysis was achieved by categorizing the identified units and interpretations into sub-themes, relational themes, and eventually meta-themes. This was followed by a summarisation of the findings into simpler, easy-to-understand sentences and finally these were synthesized and interpreted to capture the data that was contextually comparable to the original transcripts, notes and recordings. The team had several meetings to address and agree any discrepancies in understanding or coding to arrive at a unified report. No coding software was applied for data analysis, nor was participant feedback sought on the final interpretation of their responses.

### Ethical consideration

Ethical approval was obtained from the Health Research Ethics Committee (HREC) of the Lagos University Teaching Hospital. Permissions were also obtained from the Lagos State Ministry of Health, and the Medical Officers of Health in charge of the health facilities. The participants were duly informed of the purpose and objectives of the study and were assured of confidentiality. Written informed consent was obtained from each participant. Participation

was voluntary and withdrawal from the study was without consequences. A unique identification number was assigned to each client before the FGD, therefore their participation was anonymous.

## Results

In this study 8 FGDs, 24 KIIs and 3 IDIs were conducted. The FGD participants were all adults (age range 22–65 years), mostly female (75.0%), male-female ratio of 1:3. The list of all the participants is presented in Table 1.

### Client perspectives on the COVID-19 vaccine rollout program in Lagos

The results of the FGD provide information on the experiences and challenges of the COVID-19 vaccine recipients regarding vaccination and their perspectives on strategies for improvement. These results are presented in 3 broad themes: i) Knowledge of COVID-19 vaccination, recipient eligibility, vaccine availability (the plan, process and progress) ii) myths/misconceptions, vaccine hesitancy iii) Benefits, adverse effects and concerns about the COVID-19 vaccine.

**Knowledge of COVID-19 vaccination, recipient eligibility, vaccine availability (the plan, process and progress).** Many of the respondents demonstrated knowledge of the COVID-19 virus stating that the vaccine can protect individuals against COVID-19 infection and reduce the morbidity and mortality associated with the infection. The clients expressed varying opinions about the planned eligibility criteria for vaccination, some stating that everyone is eligible for the vaccine while some others opined that those who are eighteen years and above were considered eligible for the vaccine, for example in **FGD 1** (Ikorodu), clients had differing beliefs on the age for eligibility, ranging from 15 years and above to twenty years and above. None of the participants were clear about the criteria for prioritizing recipients.

Regarding vaccine availability, some clients expressed their worry about the sustainability of vaccine supply. **FGD6 P4**(Female, 40 years) stated, *"The problem with this vaccination is that we are not producing it in Nigeria here if eventually this vaccine we are taking already is no more available how are we going to complete the doses*? **FGD6 P4** further stated, "*How are we going make sure to get our second dose, this is the only challenge I have noted about it, we are supposed to make our homemade vaccines, that's all."* **FGD8 P5**(Female, 35 years) worried that *"It is not everybody that will receive the vaccine."*

**Myths/Misconceptions about COVID-19 vaccine.** Some of the clients reported that myths and misconceptions surrounding the vaccine's safety and usefulness were fueled by misinformation which was a threat to the acceptance of the vaccine during the vaccine rollout, with many people alluding to the different conspiracy theories about the COVID-19 vaccine, for example, **FGD2 P5**(Male, 58 years): "*What the people are saying (is that) because the (global) population [of the people] is too much . . . they want to use (a) style to kill people, you understand, so since they cannot (find) the way to kill people they want to use, . . . injection[s] or this vaccine to kill people, they say there is nothing like. . . COVID 19*", **FGD5 P1**(Female, 40 years): *"Some will even say if you take it for the woman you won't be able to conceive any more".*

**Table 1. Study participants.**

| Participants | Number |
|---|---|
| Clients (FGD participants) | 64 |
| Frontline workers (In-depth interview participants) | 24 |
| Key officers (Key informant interview participants) | 3 |

According to the clients, the claims propagated through social media about the harmful nature of the vaccine were the major reason for vaccine hesitancy. They explained that the health educators played a good role in correcting the misconceptions during the process of community mobilization that was conducted in each LGA. Some of the clients expressed concern about the general distrust in the motives of the government regarding the vaccine and the uncertainty about the future effect of the vaccine on recipients. According to the clients, some religious leaders forbade their followers from getting the COVID-19 vaccine.

"*In this part of the world we tend to believe more in our religious leaders and I've seen, I've watched about three pastors (name not mentioned) that are saying don't get vaccinated don't take the vaccine and I know two of them that have been vaccinated yet they couldn't go back to tell the populace that they had been vaccinated and advise them to go for the vaccine.*"**FGD6 P8**(Female, 33 years).

**Benefits, adverse effects and concerns about the vaccine.**    The clients agreed to the benefit of the COVID-19 vaccine as the primary approach to protecting oneself against the infection, also stating that the vaccine reduces the burden of the disease should a person get infected, in terms of morbidity and mortality. They provided a further benefit of vaccination stating that it would help to reduce the strain on the health workers if fewer people got ill because of the vaccine, and that herd immunity would be achieved faster, **FGD4 P2**(Female, 32 years): "*this virus we are talking about is a killer and when (people) take the injection it will protect them and others in the community when the number of people that have taken the vaccine increases among them*". It was also opined that vaccination would help improve the economy because people could return to work once vaccinated. **FGD6 P1**(Male, 28 years) "*for (the) economy to be promoted people have to be vaccinated so they must be free to go out. So many people are not able to do their buying and selling, so all of such (work) was retarded by COVID-19*"

The clients mostly reported that the vaccines were completely safe, hence their willingness to receive the vaccine, **FGD5 P5**(Male, 55 years) reported, "*No disadvantage, good for everybody*". However, some concerns were expressed by some of the clients mostly about the need for personal protection with face masks and hand hygiene after taking the vaccine. **FGD2 P7** (Female, 43 years): "*I see it as a disadvantage because some people are feeling relax[ed] that so far as they have taken the injection, "I'm already protected" and so free to move with people. . . confident that, haa! so far I take the injection I don't think I (need a facemask). Even in some health facilities, I mean the medical practitioners in some areas, they too are relaxed because they have already taken the vaccine*" A few of the clients thought that for the vaccine to reach every citizen it must be made compulsory, **FGD4 P7**: "*..making It compulsory for people to get the vaccine and people that are not vaccinated should not be allowed to enter social places*".

## Perspectives of frontline healthcare workers on the COVID-19 vaccine rollout program in Lagos

The IDI results provide information on frontline workers' perspectives on the vaccine, the community response and the challenges of the vaccine rollout and suggested strategies to mitigate the challenges. These results are presented in 3 broad themes: i) The COVID-19 vaccine in the control of the pandemic ii) Barriers to the success of the vaccination programme iii) Frontline workers' evaluation of the plan, process and progress

**The COVID-19 vaccine in the control of the pandemic.**    The health workers, all working directly with the Lagos state COVID-19 vaccination programme, expressed satisfaction with

the provision of the vaccine for the protection of all including the frontline workers such as themselves. However, **IDI 1** expressed concerns saying, *"I think the disadvantage [of being vaccinated] is that some people that are vaccinated believe they are fully covered 100%, they don't want to use their face masks anymore, they don't want to wash their hands regularly, they don't want to use their hand sanitizer, they believe they can just go everywhere without face mask"*, given that hand hygiene and face coverings are required even after vaccination for the control of the pandemic. On the other hand, **IDI 4** expressed dissatisfaction with the vaccine saying, *"The disadvantage that I can talk about for now is that it's not a 100% assurance that when you are being vaccinated against the COVID-19 you cannot be infected again."*

**Barriers to the success of the vaccination programme.** The healthcare providers discussed their concerns about a successful vaccination rollout, stating that the vaccine uptake may be especially vulnerable to myths and misconceptions surrounding the vaccine propagated through social media. *IDI 21 said, "Rumour mongering is one of the most serious barriers to mass vaccination, an important aspect of this is that we are not educating the immediate community"*, and further said, *"some say it will make them impotent, others say it will make them not see very well, kill them, kill pregnant women or their babies in the womb"*. The health workers blamed misinformation for the very frequently seen behaviour they perceived as vaccine hesitancy, stating that correct information needed to be provided to individuals as a matter of urgency if vaccine coverage is to be achieved.

**Frontline workers' evaluation of the plan, process and progress.** The scoring of the vaccination programme, when the frontline workers were asked as a general question to gauge their view, was mostly above average. **IDI 18** gave a good score saying, *"[it was] A smooth process. Out of 5 I will put it [the score] between three and a half and four"*, while **IDI 19** gave an average rating but found the process favourable saying, *"A bit average because I like the process"*. Commending the government of Nigeria and the state, the frontline workers noted that the provision of the vaccines to all the designated centres and the security at the vaccination sites strengthened the process.

> *"Security-wise, Government really did well, at every post we have policemen"*. **IDI 10**

> *"The first phase is like a plus for our Government"*. **IDI 14**

> *"I am thanking the government, they should keep it up"*. **IDI 16**

In terms of logistics, some of the frontline workers expressed dissatisfaction with the planning.

> *"I think we didn't pay enough attention to the logistics that were going to be involved, when I say the logistics am talking about down from the issue of the cotton wool that will be used, the gloves that will be used because we were supplied these things [not supplied enough of these things]. The main thing that you practically have in abundance is the vaccine itself, with the cards and bar codes and of course needles and syringes."* **ID1 17**

> *"We started the first phase and the first dose had an extension that wasn't planned for, in not planning for that extension it also cost a lot of issues in terms of logistics and even the implementers who were to carry out the work for the extra ten days were not planned for."* **IDI 17**

## Perspectives of administrative providers on the COVID-19 vaccine rollout program in Lagos

The KII results focus on the provider (healthcare administrators) perspectives surrounding the COVID-19 vaccine in Lagos. These are presented in 3 broad themes: i) The COVID-19

vaccination rollout plan and process ii) Partners, funding and logistics iii) Progress; uptake, barriers and challenges

**The COVID-19 vaccination rollout plan and process.** *The plan.* The key informants outlined the vaccination plan as earlier detailed but highlighted the strengths as follows;

Training: The providers explained that the COVID-19 vaccination rollout in Lagos State started with the setting up of a committee made up of stakeholders in the health sector, tasked with capacity building to support the vaccine rollout through training of health workers. This process was initiated by a 'Train the trainers' exercise followed by a step-down training at each LGA, **KII2**: "*We had a training apart from the national training of trainers that was conducted for three days we also had the state level training, (at the) state level the NTOT, that is the national training of trainers, (it) is actually meant for key program officers in the state that will act as . . . facilitators for state. . .! state-level training so we also had the state level training for another three days where the people at the LGAs level that is the key implementers were trained*".

Four-phased rollout: It was explained that the COVID-19 vaccination was designed to be in 4 phases according to the vaccine rollout plan.

The TEACH Strategy: The key informants all understood the TEACH strategy which was adopted in the planning of the vaccine rollout in Nigeria and Lagos State, recounting that all the elements of the strategy were rightly implemented concurrently by the providers, **KII2**: "*all these things are expected to go pari passu with one another. It is not (that) you do one phase before the next one*".

Super-sites: The key informants defined a super-site as a COVID-19 vaccination center created in each LGA to address the challenge of clients who had relocated due to work, change of residence, or any other reason who were yet to receive the second dose of the vaccine. They opined that clients' details were easily traced because of the DHIS2 client data platform that was provided, **KII2**: "*I[t']s the best because worldwide, wherever you get to, all they need to do is to just capture your (bar) code and they get the information concerning that person. Going forward I will like to suggest that anything that has to do with health I think this process is the best. . .*"

*The process.* The providers expressed mixed levels of satisfaction with the COVID-19 vaccination process, dissatisfaction mostly stemming from the internet network and electrical power issues, **KII 1** saying, "*COVID-19 vaccination [process] does not have any problem*". According to the key informant, the process was strengthened overall by;

Daily review meetings: These feedback meetings were reported to be conducted at the LGA, state and national levels to review the vaccination process and address challenges.

Electronic registration on the DHIS2 Platform: The key informants reported that the server was a formidable electronic registration technology designed for use in the vaccination process, explaining that once the biodata of the clients was recorded, a unique identification number and a QR code were generated and recorded in the vaccination record card given to the vaccine recipient while a tear back counterfoil card was retained at the clinic for record purposes. They believed that the process was innovative though inefficient due to the problems with the internet network, resulting in delays and prolonged waiting times at the vaccination centres, **KII 3**: "*And we see that, . . . the client(s) weren't even sure if (when) the whole process during the e-registration was successful*". Manual registration of vaccine recipients was reported as the temporary measure taken to mitigate the internet network server problems.

Personnel/implementers: One key informant reported that the health workers facilitated the process **KII3**: "*Well! It's been orderly like if you go to,. . . the site at . . ., you will see where we have,. . . various officers that are involved*". It was reported that trained personnel formed the teams working in the COVID-19 vaccination rollout exercise. Each team consisted of a supervisor, a vaccinator, recorder, crowd mobilizer, town announcer and security personnel.

However, the constituents of a team were said to have been reviewed to include two recorders and no town announcers, given that the role of an announcer was no longer required after a few weeks, while independent monitors gave daily reports at evening review meetings. it was further reported that the strength in the process was in the Local Government Area vaccine teams which consisted of the Local Immunization Officer (LIO), the Disease Surveillance and Notification Officer (DSNO), the Cold Chain Officer (CCO), the Apex Nurses and the Medical officer of Health who were in charge of supervising and monitoring the vaccination process. The partners such as the WHO representatives, National Agency for Food & Drug Administration and Control (NAFDAC) and the State Technical Facilitators were reported to have also been deployed to each Local Government Area to assist in training, monitoring and supervision of the COVID-19 vaccination process to ensure its success.

Channels of reporting adverse effects following immunization (AEFI): It was reported that to observe for AEFI after immunization clients were required to wait at the vaccination centre for 30 minutes observation and advised to call the number on the vaccination card or visit the clinic for treatment if they observed any side effects after leaving the vaccination centre. According to the key informants, data on AEFI were collected through the line-listing database developed to monitor side effects, while the Med safety app [28–30] was used for remote line-listing.

**Partners, funding and logistics.**    *Partners.* The key informants reported that some partners also collaborated with the NPHCDA and the State Government to ensure a successful vaccination exercise, stating that the role of the collaborators was to support the COVID-19 vaccination programme with logistics, funding and monitoring. Some of the partners listed were the WHO, UNICEF, NAFDAC, etc.

> "*The partners WHO, UNICEF, CHEA and all the other partners, they are also involved in supporting the process.*" **KII 2**

*Funding.* The key informants discussed the need for more funding to support the programme in addition to the current funding from the NPHCDA and the Lagos State Government, KII 1: "*We also have to mobilize funds that will be used for (supporting) the vaccination (with COVID-19 vaccine)*".

*Logistics.* The key informants reported that the tools, materials, and kits needed for the vaccination process were supplied to all the vaccination sites in Lagos State. **KII 3**: "*The vaccination sites have all the necessary materials to be used for [the] vaccination*". These materials included face masks, hand sanitizers, gloves, cotton wool, vaccination cards, registration forms and registers, portable tablet computers and android phones used to facilitate the registration of clients. It was reported that the vaccines were stored and supplied to the sites fully applying the cold chain system, which they described as intact, **KII 2**: "*[The] vaccine was brought to the state cold store and it was saved at the approved appropriate temperature*", and adequate provision was made for security at these sites. Vaccine Accountability Officers, who function to monitor and ensure the cold chain of vaccines was maintained by each vaccination team, were trained to monitor and ensure vaccine safety and were reported to have been at every site. It was reported that through community support from the Local Government, some vaccination sites provided tents and chairs for clients waiting to be vaccinated, an action they considered commendable.

**Progress; uptake, barriers and challenges.**    Most of the key informants were positive about the vaccination progress and were optimistic about coverage provided that the planning, logistics, supplies and public enlightenment required were optimal. **KII 2** suggested "*improvement of the vaccination process*". **KII 3** suggested adequate and timely payment of the salaries

of the workers. The bottom-up approach was recommended for consideration in future vaccination planning. Recommendations were made for incorporating the COVID-19 vaccination programme into the routine immunization programme already established at each PHC to reduce the cost of logistics and ensure the much-needed wider coverage. Also recommended were, local COVID-19 vaccine manufacture because of the large population in Nigeria, more vaccination teams and sites, gaining the support of partners and non-governmental organizations (NGOs) and Public-Private Partnership through the participation of the private health facilities in the delivery of the COVID-19 vaccination, **KII 1** *"in some countries they have the public-private partnership (that can aid) the vaccination (process) and then (the) citizens pay (a) token to get their vaccines because the government cannot afford (to do I all). (It is) one thing for the government to receive the vaccine from the partners . . . but the cost of even the logistics of carrying out the vaccination is even more (than) the cost of the vaccine itself, and government cannot continue (to provide vaccines)"*.

*Uptake*. Vaccine uptake was reported as extremely low in some centres. The poor turnout for vaccination was perceived by key informants to be linked to poor awareness. They noted that the widespread vaccine hesitancy was due to rumors and misinformation surrounding the vaccine. **KII 2:** *"they read the jargon here (on social media), "haa! I don't want to take o, haa!" So that is why you find out that even at the beginning of the vaccination majority of people were skeptical. People were thinking, "let us look out (for) how many people will die after taking the vaccine", so after waiting for about one or two months they see that nobody is dying, rather people are living in good health so the majority are now registering (to take the vaccine)"*. The key informants recommended that public enlightenment through vaccination campaigns to aid the progress of the vaccine rollout was the surest approach to improving vaccine uptake using social media and the time-tested traditional broadcast media through rallies, market campaigns, billboards, fliers, talk shows, etc. were recommended. The information being broadcast should aim to debunk the wrong information being propagated. The concept of Vaccine Champions was introduced by **KII 1 as** *"well-known, role models and reputable people who have received the vaccine and who will promote the vaccine uptake by encouraging people to get vaccinated"*.

*Barriers*. Aside from challenges related to planning, a few barriers notably poor public awareness, misinformation, myths and misconceptions, logistics, poor internet connectivity, poor electricity supply, lack of functional tablet computers, and in some cases, inadequate consumable materials like cotton wool, gloves, vaccination cards, QR code generation, face masks and the vaccines at some sites, and finally, poor staffing were reported.

In addition to misinformation, AEFI such as pain at the injection site, headache, weakness, excessive eating and excessive sleeping were reported as major factors leading to vaccine hesitancy and poor uptake even though only a few serious complications like fainting and other situations requiring hospitalization were reported.

*Challenges*. The challenges associated with the COVID 19 vaccination were termed social, psychological and economic.

Social challenge: A key informant reported the stigma associated with receiving the vaccine caused clients to receive the vaccine in secret because they feared rejection from their families. **KII 3:** *"Rumor is being spread that for those who received (the vaccine) will die and all that, it will also affect their social life", "(a client said) they received (a phone) call from a cousin saying that they are going to die because they have been vaccinated"*. Stigmatization was viewed as disruptive to the vaccination rollout process.

Psychological challenge: A patient-related challenge to the entire COVID-19 response highlighted by key informants was that despite contrary information, people who were vaccinated let their guards down in terms of observing COVID-19 protocols such as hand washing,

social distancing, wearing of face masks, etc. Also, some individuals delayed receiving the second dose of the vaccine because they were psychologically unprepared for it, **KII 2**: *"some usually have [a] fever and then pain at the site of injection and then we also have reports of people throwing up, eating more than expected, sleeping and then headache and so on and so some are afraid of the second dose"*, owing to the side effects they experienced with the first dose.

Economic challenge: A key informant expressed concern over the seeming imminent requirement of vaccination for travel and tourism and the sustainability of the COVID-19 vaccination exercise because of the high cost associated with the vaccination rollout process, KII 1: *"I've told you about how it [the COVID-19 vaccine roll-out process] affects us socially and economically and that is why the government cannot continue to support and to spend money on it because it's affecting other spheres of life, (it) is affecting the other sectors, for example, we have huge unemployment (due to COVID-19). People who were employed before (have) lost their jobs, and (it) is very difficult for people to even earn a living so (the) government (has to) put money in other sectors so that the economy can come up"*, also stating that proper planning informed by adequate monitoring and evaluation is needed to ensure the success of mass vaccination.

The perspectives of the clients and providers of COVID-19 vaccination about the vaccine rollout plan, process and progress, as well as the suggested strategies to address the identified challenges, gathered through the FGDs, IDIs and KIIs are as summarised in Table 2.

## Discussion

This study sought to examine the challenges of COVID-19 vaccination rollout in Lagos, Nigeria from the point of view of clients and healthcare providers. The complexities of the factors that have affected the vaccine rollout are demonstrated in the findings, bringing to light the challenges observed by the vaccine recipients, the frontline health workers and the health administrators in Lagos State. The challenges were related to vaccine recipient eligibility, vaccine availability, myths/misconceptions, conspiracy theories, concerns about the degree of protection offered by the vaccine, vaccine hesitancy, uptake and coverage, completing vaccine doses, the use of electronic data capturing, the role of international partners, funding and logistics. The primary findings of this study align with the challenges, barriers and facilitators reported in the 49 African countries that started rolling out COVID-19 vaccines in early 2021 [6]. Several previous studies in various parts of the world and Africa have identified challenges both systemic and individualized including vaccine hesitancy and inadequate fact-based information on the benefits and risks of the COVID-19 vaccine. Some studies have suggested that vaccine hesitancy may be particularly prevalent in low-income countries like Nigeria [31], where our study was conducted, precipitated by factors such as distrust in vaccines [32]. Unfortunately for countries in Africa from December 2020, even before doses of the COVID-19 vaccines became available in Africa, the global movement against the vaccine had created a breeding ground for vaccine hesitancy as was the case in Nigeria [33]. This growing concern remains a threat to the vaccine rollout even after several months into the process. One study found that fear is significantly correlated with the lack of accurate vaccination knowledge leading to a poor perception of the importance of vaccination and a lower likelihood of vaccine uptake [34].

Studies also reported challenges related to the operational plan for vaccine rollout [35,36], along with vaccine shortages, poor logistics and supplies [37]. However, poor internet network was the most limiting challenge reported in our study causing poor linkages between internet servers and external facilities, hence a resort to paper registration. The electronic registration for COVID-19 vaccination is a novel technology in the immunization history of Nigeria, therefore the challenges experienced during the first phase of the vaccination process are such that can be addressed and improved upon in subsequent vaccination exercises.

**Table 2. Summary of client and provider perspectives on the challenges of the COVID-19 vaccine rollout plan, process and progress and strategies to address the barriers to successful vaccine rollout.**

| Thematic area of reported Challenges | Challenges of the COVID-19 vaccine rollout | Strategies to address the barriers to successful vaccine rollout |
|---|---|---|
| **Client perspectives** | | |
| Recipient eligibility | Confusion about the categories of individuals eligible to receive the vaccine. | Community-level enlightenment campaigns and community mobilization. |
| Vaccine availability | Too few doses were available for the huge population of people in Nigeria. | Local vaccine manufacturing. |
| Myths/misconceptions | Misleading information about the vaccine propagated especially on social media leading to vaccine hesitancy, and apprehension about the second dose. | The deliberate provision of adequate and correct information about the vaccine. Making the vaccine compulsory. |
| Conspiracy theories | Distrust in the health system and the government causes people to ignore vaccination. | Community enlightenment campaigns, transparency in governance, leadership with integrity |
| **Provider perspectives** • **Frontline workers** | | |
| The degree of protection offered by the vaccine | Vaccination creates a false sense of security from the infection so people no longer want to apply safety measures. | Continuous health education |
| Vaccine uptake | Poor vaccine uptake due to myths and misconceptions among members of the community | Urgent correct public enlightenment |
| Frontline workers' evaluation of the plan, process and progress | Participation is limited by poor logistics. Insufficient and inconsistent supply of consumables. Unplanned changes and the extension of the first phase disrupted the original plan. | Better planning, backed by correct and accurate data. |
| • **Administrators** | | |
| Completing vaccination doses | Clients are at risk of being lost to follow-up for the second dose of the vaccine. | Strengthening and creating more supersites. |
| Use of electronic data capturing | Poor internet data network and electrical power supply resulting in delays and prolonged waiting times at the vaccination centres. | Developing an alternative and appropriate technology that does not depend on the inconsistent power supply in the country. Improvement in internet network in the country as a whole. |
| Partners, funding and logistics | The limitation in the logistics related to inadequate funding | More collaborations and funding to support the program |
| Evaluation of the progress; uptake, barriers and challenges | Inadequate supply of consumable materials like syringes, a reflection of the need for collaboration. | The bottom-up approach is to be applied in planning so that the steps taken are more locally appropriate. Future COVID-19 vaccine integration into the routine immunization programme to reduce the cost and ensure coverage. |
| Vaccine uptake and coverage | Vaccine hesitancy due to poor vaccination information. The stigmatization of vaccine recipients Psychological challenges associated with receiving the vaccine. Poor internet connectivity, poor electricity supply, lack of functional tablet computers, inadequate consumable materials like cotton wool, gloves, vaccination cards, QR code generation, face masks and poor and insufficient staffing. | Targeted vaccination campaigns with social media and broadcast media sensitization to debunk myths. 'Vaccine champions' to address stigma. Counseling before and after vaccination. Rigorous monitoring and evaluation of the vaccination process for future planning |

Having adequate knowledge about vaccination has the potential to increase vaccination uptake directly by addressing misconceptions and providing clarity on the vaccination schedule or indirectly by helping eliminate the fear of vaccines. One of the most common vaccine-related misconceptions is that the benefits of vaccination are minimal compared to the often-exaggerated risks and that vaccination is inferior to natural immunity acquired from surviving a disease [38]. This study however reports concerns about misconceptions from misleading information about the side effects of the vaccine, propagated especially on social media,

leading to vaccine hesitancy and apprehension about taking the second dose of the vaccine. These misconceptions have led to a distrust in the government and the health system causing people to ignore vaccination, though the turnout for vaccination was very impressive at some vaccination sites. However due to misinformation and misconceptions similar to that found in other countries that are involved in vaccine rollout [39,40], the vaccination process was somewhat limited such that in some centres turnouts were particularly poor. Among the factors that affect vaccine decision-making, cultural, social and vaccine-specific factors play a minimal role when people do not trust the vaccines available [41]. The term vaccine confidence refers to the trust that clients and providers have in vaccines, the vaccination process, the health workers who administer the vaccines, vaccine production and policies [42]. It stems from an understanding of the importance of building trust in COVID-19 vaccines. Vaccine uptake is improved with increased vaccine confidence [43] leading to reduced COVID -19 morbidity and mortality. To promote vaccine confidence the risk communication about the vaccine should be clear [44] as recommended by participants, such as in their local language or using tailored language that they can easily understand [45]. The local language can be used to address the myths, misconceptions, rumors, misinformation and disinformation surrounding the COVID-19 vaccine.

Other studies investigating predictors of vaccination uptake, and factors associated with poor vaccination coverage found that the perceived severity in terms of the high morbidity and mortality of a disease such as COVID-19 may reduce vaccine hesitancy [46], demonstrating that vaccine uptake is highest before and during a pandemic, and immediately after a new vaccine is available to control the pandemic [47]. This knowledge should inform the action of the healthcare community to capitalize on this predictable early COVID-19 vaccine enthusiasm from the public to proceed with vaccine rollout using a well-organized, rapid vaccine distribution plan [48]. In Lagos State, full vaccination was facilitated by mitigating dropouts, using 'supersites' created in each of the Local Government Areas of the state to address the issue of second dose vaccination.

Aside from health system-centered challenges, clients reported barriers to vaccine uptake, notably, AEFIs. The vaccine's side effects were mostly minor, for example, the more frequently reported symptoms like fever, pain at the injection site, headaches, excessive sleeping, excessive eating and weakness, none lasting beyond forty-eight hours. While more serious side effects were less frequently reported, a few cases of severe AEFIs requiring hospital care were documented in some centers including fainting, collapsing and difficulty in breathing.

Another challenge discussed among study participants is that the COVID-19 vaccine doses may remain perpetually insufficient given Nigeria's dense population. However, vaccine rollout can progress to reach all citizens with good planning, adequate funding, an efficient vaccine cold chain system and logistics that take into cognizance the differences at the local, state and national levels.

The administrators recommended that the citizens are carried along and their opinions included in the vaccination planning process describing a bottom-to-top approach. This is effective as shown in a vaccine study that shows that the bottom-up strategy outperforms the top-down strategy in community vaccine acceptance and involvement in vaccine programmes [49]. This study revealed that the vaccination process can also be improved upon with the provision of more healthcare teams, more vaccination sites and better logistics.

One approach that may be important is that COVID-19 vaccination should be incorporated into the routine immunization programmes that already exist in Nigeria. This would promote wider coverage and decrease the cost of vaccination and therefore, the funding needed by resource-constrained countries like ours. Furthermore, integrating COVID-19 vaccination with routine immunization would potentially prevent collateral disruptions in other essential

services like childhood immunization, antenatal care, etc., which are offered at the primary health care facilities, the sites for vaccination in the State.

Although the majority of the participants in this study were not in support of making vaccination mandatory a few participants suggested enforcing it. This opinion has been expressed by several writers, some specifying their recommendations to several priority groups like health workers and care home workers [50]. Participants suggested that public-private partnerships can be used to ensure a sustainable vaccination process because the government may not be able to bear the cost of vaccine delivery in the long run. This opinion may have been influenced by the numerous examples of the successful interface of the government and private organizations in healthcare provision in many settings in Nigeria [51,52] and globally [53,54]. COVID vaccine rollout would be similar to these examples in the sense that the limitations of poor funding would be eliminated through these partnerships. Nigeria and indeed other countries can learn from our findings to leverage available resources, and ensure that development is ongoing and can compete with other nations in the world, placing priority on measures such as education and literacy.

## Conclusions

The vaccination process though without major threats was limited by internet network problems which affected the electronic registration. Misinformation and misconceptions were recognized as some of the barriers which created vaccine hesitancy. The AEFIs reported were mild. The participants recommended continuous public enlightenment to provide fact-based vaccination information to help build trust and confidence in the COVID-19 vaccine. Valuable lessons are emerging from the rollout process including the importance of the bottom-up approach in planning, more teams, more vaccination sites and more vacccines to ensure a wider reach and coverage.

## Acknowledgments

All respondents for their participation in the study.

## Author Contributions

**Conceptualization:** Oluchi Kanma-Okafor, Yetunde Odusolu, Akin Abayomi, Faisal Shuaib, Moji Adeyeye, Ibrahim Mustapha, Segun Ogboye, Dayo Lajide, Hussein Abdur-Razzaq, Ukamaka Okafor, Uchenna Elemuwa, Akin Osibogun.

**Data curation:** Oluchi Kanma-Okafor, Yetunde Odusolu, Akin Abayomi, Faisal Shuaib, Moji Adeyeye, Ibrahim Mustapha, Segun Ogboye, Dayo Lajide, Hussein Abdur-Razzaq, Ukamaka Okafor, Uchenna Elemuwa, Akin Osibogun.

**Formal analysis:** Oluchi Kanma-Okafor, Yetunde Odusolu, Ibrahim Mustapha, Hussein Abdur-Razzaq.

**Funding acquisition:** Yetunde Odusolu, Akin Abayomi, Faisal Shuaib, Moji Adeyeye, Ibrahim Mustapha, Segun Ogboye, Dayo Lajide, Hussein Abdur-Razzaq, Ukamaka Okafor, Uchenna Elemuwa, Akin Osibogun.

**Investigation:** Oluchi Kanma-Okafor, Yetunde Odusolu, Akin Abayomi, Faisal Shuaib, Moji Adeyeye, Ibrahim Mustapha, Segun Ogboye, Dayo Lajide, Hussein Abdur-Razzaq, Ukamaka Okafor, Uchenna Elemuwa, Akin Osibogun.

**Methodology:** Oluchi Kanma-Okafor, Yetunde Odusolu, Akin Abayomi, Faisal Shuaib, Moji Adeyeye, Ibrahim Mustapha, Segun Ogboye, Dayo Lajide, Hussein Abdur-Razzaq, Ukamaka Okafor, Uchenna Elemuwa, Akin Osibogun.

**Project administration:** Oluchi Kanma-Okafor, Yetunde Odusolu, Akin Abayomi, Faisal Shuaib, Moji Adeyeye, Ibrahim Mustapha, Segun Ogboye, Dayo Lajide, Hussein Abdur-Razzaq, Ukamaka Okafor, Uchenna Elemuwa, Akin Osibogun.

**Resources:** Akin Abayomi, Faisal Shuaib, Moji Adeyeye, Ibrahim Mustapha, Segun Ogboye, Dayo Lajide, Hussein Abdur-Razzaq, Ukamaka Okafor, Uchenna Elemuwa, Akin Osibogun.

**Supervision:** Oluchi Kanma-Okafor, Akin Osibogun.

**Validation:** Akin Abayomi, Moji Adeyeye, Segun Ogboye, Dayo Lajide, Hussein Abdur-Razzaq, Ukamaka Okafor, Uchenna Elemuwa, Akin Osibogun.

**Visualization:** Oluchi Kanma-Okafor, Yetunde Odusolu.

**Writing – original draft:** Oluchi Kanma-Okafor, Yetunde Odusolu, Faisal Shuaib, Akin Osibogun.

**Writing – review & editing:** Oluchi Kanma-Okafor, Yetunde Odusolu, Akin Abayomi, Faisal Shuaib, Moji Adeyeye, Ibrahim Mustapha, Segun Ogboye, Dayo Lajide, Hussein Abdur-Razzaq, Ukamaka Okafor, Uchenna Elemuwa, Akin Osibogun.

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
