## [Decision Letter · Decision Letter 0]

23 Feb 2022

PGPH-D-21-01166

A qualitative analysis of the COVID-19 vaccination rollout in Lagos, Nigeria: Client and provider perspectives on the plan, the process and the progress

Dear Dr. Oluchi Joan Kanma-Okafor,

Thank you for submitting your manuscript to PLOS Global Public Health. After careful consideration, we feel that it has merit but does not fully meet PLOS Global Public Health’s publication criteria as it currently stands. Therefore, we invite you to submit a revised version of the manuscript that addresses the points raised during the review process.

We look forward to receiving your revised manuscript.

Kind regards,

Muhammed Olanrewaju Afolabi, MD, MPH, PhD

Academic Editor

JOURNAL REQUIREMENTS:

1. Please update the completed 'Competing Interests' statement, including any COIs declared by your co-authors. If you have no competing interests to declare, please state "The authors have declared that no competing interests exist".

2. In the online submission form, you indicated that "Data is available on request but cannot be made publicly available for ethical reasons, to protect the respondents.". All PLOS journals now require all data underlying the findings described in their manuscript to be freely available to other researchers, either 1. In a public repository, 2. Within the manuscript itself, or 3. Uploaded as supplementary information.

3. Please amend your detailed Financial Disclosure statement. This is published with the article, therefore should be completed in full sentences and contain the exact wording you wish to be published.

i) Please include all sources of funding (financial or material support) for your study. List the grants (with grant number) or organizations (with url) that supported your study, including funding received from your institution. 

ii). State the initials, alongside each funding source, of each author to receive each grant.

iii). State what role the funders took in the study. If the funders had no role in your study, please state: “The funders had no role in study design, data collection and analysis, decision to publish, or preparation of the manuscript.”

4. Please ensure that the funders and grant numbers match between the Financial Disclosure field and the Funding Information tab in your submission form. Note that the funders must be provided in the same order in both places as well.

REVIEWERS COMMENTS:

Reviewer #1:

Overall, this paper focuses on the COVID-19 vaccination rollout in Lagos from the perspectives of vaccine recipients, HCWs, and government officers. Overall, the authors did a great job providing and presenting a lot of data on the rollout; I particularly like the inclusion of different perspectives. Some of the language can be cleaned up throughout, but overall, I think it’s a paper the authors should be proud of and is one that will benefit future vaccine rollouts in Nigeria and elsewhere.

Some detailed comments are provided:

Introduction

- Line 80: Provide a little more detail – perhaps the number of vaccines currently produced (can say “at the time of writing”) to provide context of scale

- Lines 80-84: Can you be more specific? Is this clinical data? Consider breaking the sentence into two separate ones for clarity.

- Line 84: were the blood clots discredited, or was it reports that it was unsafe? Please clarify.

- Line 85: Is it possible to be more specific than “Africa”? Maybe “Vaccination against COVID-19 has been slower in countries relying on doses from COVAX than in other places” or something like that.

- Lines 88-89: This is a very important point, but the sentence is unclear. What proportions are you speaking of?

- Line 90: An example of an exemplary rollout would be helpful.

- Lines 91-95: This sentence needs some citation(s), as there have been several reports about why the rollouts/administration may have been delayed – including, but not listed here, countries receiving vaccines right before they were to expire.

- Line 96: Which vaccine was provided?

- Line 106: Did Lagos State receive their share immediately?

- Line 124: When did the vaccine become available to all adults?

- COME BACK TO AIM/LINE 125

Methods

- Overall, the divisions and sub-divisions are a bit unclear. Consider combining some sections or making note of how sub-sections may fit into broader sections (using numbers and/or letters).

- Line 134: Lagos city or Lagos state?

- TOO MUCH INFO? Perhaps link to rollout more and cut the rest… And then connect with study sites

- Lines 158-162: What does it mean to be formally approached? How was that different than approaching vaccine recipients?

- Data collection tools and techniques: I’m a little confused by “qualitative inquiry.” You mean the qualitative data were collected using a guide, correct? Was this guide structured or semi-structured? Please provide a little more detail on the key topics addressed in the guides and how they differed between IDIs and KIIs. Same can be said for the FGDs – please provide a little information on key topics covered.

o Where FGDs held the same day as the recruitment of participants? Or did participants return on a separate day?

- Line 180: The number of IDIs, etc. carried out should be included in the results section. Also, if there were 8 sites and 4 IDIs were carried out at each site, why were 24 conducted (rather than 32)? 24 is still a lot of IDIs, but please comment if there were decliners or a reason why the numbers do not match.

- Lines 183-186: Is there a way to not include their titles? I imagine this makes the KII participants easy to identify… Please be more general in how they are referred to.

- Lines 191-192: Please clarify. “Transcribed verbatim” means word-for-word, so by the rest of the sentence, do you mean that the other notes (e.g. non-verbal cues, etc.) were included into the transcripts as well?

- Line 196: “manuscripts” should be changed to “transcripts”

- Lines 199-207: There is a lot of detail here that I believe can be written/edited in a way that is more succinct and clearer to the reader.

o If no coding software was used, did the authors use Word or Excel?

Results

- The number of KIIs, IDIs, FGDs should be included at the beginning of the results section. Here you can also include some summarized information on the participants (gender, age, etc.)

o Were men and women included in the same FGDs?

o What is the gender breakdown of the IDIs and KIIs?

o I’m still worried that by providing their titles, the identity of the KIIs is easy to figure out.

- Lines 252-254: did the participants themselves have concerns about the government’s motives or the uncertainty of the effect, or was it what they had heard from others? If they had these concerns, what led them to still get vaccinated?

- In this section (myths/misconceptions), it would be helpful to have a specific example or two. Participants seem to have heard of the myths (did they mention any specific myths?) but still got vaccinated. Did anyone say why that was?

o OK, this is mentioned in the next section. Was it the same respondents?

- Lines 303-305: Was this hesitancy the HCWs had seen while working in the clinic – was it directly observed by them? Or was this more of an explanation for wider vaccine hesitancy?

- Lines 308-311: Were participants asked to score the programme during the IDIs? Was this a formal assessment, or was it part of the interview questions? If it was formal, please say more on that and include it in the methods section.

- Line 323: should there be “not” in that sentence?

- Lines 339-351: Since the steps are already mentioned in the intro, I think this section can be cut. Or, the focus should not be on repeating the steps, but on the KII participants’ views on how this went, barriers/successes, etc.

- Section from KIIs in general, lines 331-461: I have a few concerns about this section.

o First, there is a lot of space given to what was only 3 KIIs in comparison to the IDIs and FGDs. Much of the first part is just a summary of the process that occurred, much of which is included in the intro. I think what is most relevant for the study can stay in the intro and then the rest can be removed and instead placed in a figure or text box, possibly to be included in the annex or as supplementary information.

o The parts that bring up the KII participants’ opinions on what went well and what can be improved upon are good and could be elaborated on – especially as their comments relate and compare to what the IDI participants said.

o Lines 423-425: is this the opinion of the participants? As it reads, it sounds almost like they are summarizing a study they did.

o I’m not sure what is different between “barriers” and “challenges.”

- In general, the sub-sections of the Results all follow the same order of plan, process, and progress for each group of participants separately. Something to consider would be blending the results from the different participants so they can be compared and contrasted as it relates to this overall main outline (plan, process, progress). This would keep make the results more succinct, allow for comparison between different groups of respondents, and make the sections and sub-sections easier to follow.

Discussion

- Overall, there is a lot of information in the discussion, but I feel it’s typically very broad and general. I think this section would benefit a lot from clear examples in the literature and how those examples link directly with the data presented in the results.

- Lines 480-481: Sentence needs to be written more clearly.

- Lines 502-507: Are there other reasons why certain sites had better turnout than others? Was it only do to misinformation, or are there structural factors that may play a role?

- Lines 507-508: Citation?

- Lines 513-516: Can you be more specific? Perhaps include more of the information participants gave regarding how or what they want included in vaccine messages and how the rollout may address those issues in the future.

- Lines 517-520: I’m not sure I understand this sentence… How does high morbidity/mortality lead to a reduction in vaccine hesitancy? Also, this is a rather old source – is there anything newer that addresses this issue, particularly as it may relate to COVID?

- Lines 526-531: Is it the AEFIs themselves, or people’s perceptions of the AEFIs that is most important for vaccine uptake?

- Lines 536-541: I find that participants wanted a more bottom-up approach interesting. Was this something that the KII participants discussed too? Is there a way to compare/contrast this desire with the plans of the rollout?

- Lines 542-548: Have there been any studies about incorporating other vaccines into routine immunization?

- Lines 552-556: Can you provide an example of the use of these types of partnerships? How would COVID vaccine rollout be similar or different from these examples? Are there importance lessons that can be applied?

- Overall, how can other countries learn from Nigeria? This study gives details on many challenges and lessons learnt – how can other countries benefit from this information? How can Nigeria benefit from this information in the future?

Reviewer #2:

This paper reports on a qualitative study that assessed the plan, process, and progress of the COVID-19 vaccination program in Lagos State Nigeria. Using in-depth interviews and focus group discussions, the authors highlight, from the perspectives of vaccine recipients and healthcare workers, the challenges to the COVID-19 vaccine implementation program in Lagos State including issues around vaccine hesitancy and logistics. Factors that contributed to the success of the program are also presented. The study is novel and has interesting and important findings that could be used to inform future vaccine programs in Lagos and Nigeria generally

The presentation of the paper however needs to be improved in terms of structure, additional information in the Method’s section, and support (quotes from IDIs and FGDs) for some of the key results. My comments are as follows:

• Going through the results and certain sections of the abstract, it became less and less clear if the manuscript is reporting on the outcome of research or a monitoring and evaluation exercise. Perhaps this is because of the way the results are presented (especially the choice of words). For example, participants agreed, commending the government of Nigeria, and detailed descriptions of the actual roll out program instead of presenting the perspectives of the different stakeholder groups without the authors qualifying them as: “genuine” “successful, “unfounded claims” etc. Also, where the authors present the results of IDIs with healthcare workers and key informants, there is an inclination to provide a defence to most of the challenges mentioned by the interviewees. That may be okay, if those reasons are provided in the discussion section for example… although in the interviews, participants mentioned that XXX was a major logistic-related challenge, from our knowledge and experience of the roll out program, this challenge may have been due to XX and was resolved by XX.

• Following up from the point above, it is important for the authors to describe their positionality in the context of this research. For example, were some of them involved in the vaccine roll out program and if so in what capacity and how this may have influenced the results or discussions? This will give researchers some perspectives on the overall narrative and arguments advanced in the paper.

• The methods section needs more information:

• Line 159: vaccine rollout process or vaccine roll out program?

• Line 160: It seems to me like two sampling strategies were used. Purposive (frontline workers) and convenience (clients/vaccine recipients).

•Line 161: Participants were formally approached- What does this mean? -A description of the process and why that was necessary will provide more clarity

• Line 164: Delete “For the qualitative inquiry”. The title already states that it is a qualitative study. Also, only qualitative methods were used

• Line 164: Moderator’s guide: What was the nature of this guide? Semi structured, open ended or close ended? What type of questions were included and how was it developed?

• Line 165: State the number of IDIs and KII. What was the difference between the two? was it in the method or in the stakeholder group interviewed? If in methods, then I suggest that should be described. If it is in the stakeholder group, then I suggest you use IDIs across the paper.

• The methods section should have a justification of why different methods were used for the different stakeholder groups. For example, why FGDs with vaccine recipients and not IDIs. Similarly, why IDIs with frontline workers and not FGDs. Also, it needs to be stated clearly at the introduction that FGD were conducted with vaccine recipients only and IDIs with healthcare workers and key informants (Administrators?) only. Otherwise, it becomes confusing who was involved in what, especially in the results section.

• How long were the IDIs and KIIs? This information was provided for FGDs but not the other methods.

• How did the authors arrive at the final sample size for each stakeholder group and in the case of FGDs, why 8 participants per FGDs?

• What was the overall study period? When were the FGDs conducted, was it immediately after vaccination? If yes, how were participants selected from the overall pool of persons who presented at the vaccine site on that specific day?

• What was the criteria for selecting the key officers? That is how did the team decide who was a “key informant”? for example why the state epidemiologist and not the Commissioner of Health or the Head of the Nigeria CDC?

• Line 188: Were all the Investigator part of the training or only those responsible for the conducting the FGDs, KIIs etc If not all, consider indicating the initials of those who were trained for this purpose.

• Line 195-196: What does transferability for cultural reasons mean and what impact did this have on the thematic coding and data analysis.

• Line 196: Read the manuscript (transcript?)

• Line 199: A description of the theoretical framework is important to give readers a perspective of the analysis. Is it an existing framework that has been published or used in similar studies or was it developed during the coding process by those involved in the analysis? - If so, was it developed etc. It will also be important to state if the thematic coding was inductive or deductive.

• Line 200: The sentence, “Next, by the process of indexing. This process of indexing needs to be described or referenced for the benefit of your readers.

• Line 203: What is understandable data? Either provide a description or provide a reference of what it means in the context of qualitative research.

• Line 205: Were the several meetings with the four team members involved in the coding or all the authors.

• Line 213: Purely voluntary- I suggest purely be deleted except it goes beyond voluntary.

Results:

• Table 1: Headings in Column 1 are not consistent- In one instance the method is used, in another the name of the stakeholder group. I suggest the authors use 1 across all groups, preferably the stakeholder group (Vaccine Recipients; Front line health care workers etc)

• Overall, the presentation of the results is confusing. It is presented according to the Methods. I suggest the headings should rather be defined by the different Stakeholder Groups. For example, “Perspectives of vaccine recipients on the COVID-19 vaccine rollout Program in Lagos”; “Perspectives of frontline healthcare workers on the COVID-19 vaccine rollout Program in Lagos” etc.

• Line 234: Introduce a quote from the FGDs to support the statement.

• Line 245: The theme needs to be rephrased to capture the goal of the project (COVID-19 vaccine roll in Lagos). For example, “Myths or misconceptions about COVID_19 Vaccine and the impact on the vaccine-roll out program in Lagos State”. The paper is on the vaccine roll out process and not COVId-19 vaccine hesitancy. How it leads to hesitancy can then be captured in the narrative rather than the sub-heading (a key theme)

• Line 249: A supporting quote should be inserted at the end of the sentence “…… conspiracy theories about theCOVID-19 vaccine” to support the analysis/narrative

• Line 249: Sentence starting with 'Unfounded claims'…. I suggest that as much as possible, the authors should remain value neutral in presenting the results and avoid the use of words like “unfounded” except that was used by the participants and there is a supporting quote for that. That phrasing, if necessary, can be used in the discussion section with a supporting argument on why they are considered “unfounded”. I think here is an opportunity by the authors to educate, rather than then dismiss a claim as unfounded” because such dismissals could entrench covid-19 vaccine hesitancy or resistance.

• Line 250: Participants mentioned that..….. Was this all or some, or most of the participants? This applies across the manuscript.

• Line 267. Sentence ending with “achieved faster” needs a supporting quote. Throughout the results, please provide supporting quotes for each result.

• Line The statement “It was also opined that vaccination would help improves the economy…” also requires a supporting quote

• As mentioned earlier, consider replacing In-depth Interviews with a description around the lines “Frontline Workers' Perspectives on COVID-19 Vaccines the roll out program in Lagos Nigeria”

• Why are supporting quotes from the IDIs not reported in the same way as the FGDs? The FGDs for example had the gender and age of the participants in addition to the unique participant number, while the IDIs only have the unique IDs. This needs to be consistent across the paper. Analysis in terms of age and gender are also not presented, so it is hard to tell why the authors reported the FGDs using gender and age.

• Lines 314-318: Up to three quotes are provided to support the results that HCWs commended the government, and all quotes are very similar. I suggest one quote is used for this purpose especially as in most cases, just one quote is provided for a key result- and in other cases no quotes.

• Lines 320- Kindly provide description on the specific aspect(s) of planning that came up from the interviews before providing a quote. One quote should be okay.

• Line 331: Instead of key informant interviews, consider using “Providers Perspectives on COVID-19 Vaccine Roll out in Lagos Nigeria”

• Line 332: Considering the length of this manuscript and readers may have forgotten which stakeholder groups were described as “Providers”, It is important, for the purpose of clarity, to repeat that information in this sentence - May be important to repeat it here for clarity.

• Line 339-351: I suggest the paragraph be deleted as it doesn’t form part of the key results nor adds value to the results. Same for like 357-359.

• Line 367: Please provide a supporting quote.

• Line 370-71: Why was there a need to review the process? I think that is the key information as the article reports on vaccine roll out process.

• Line 386: Issues around logistics has already been reported. Also, Lines 397-407 seems to be an evaluation of the logistics rather than the perspectives of the key informants. I therefore suggest that the paragraph be deleted.

• Lines 442: Please provide supporting quote

• Lines 443: For the entire section, please provide supporting quotes for each of the key results.

• Lines 450-453: It is not clear how failure to observe COVID-19 protocols by the already vaccinated, was a challenge to the vaccine roll out program in Lagos. That will need further explanation.

• Lines 469: The word client has only been introduced in the discussion. My guess is that this refers to vaccine recipients. However, I think, the authors should opt for one description for that stakeholder group and use it throughout the manuscript.

• Overall, the use of “participants” across the results section is confusing, I suggest it should be replaced with the title of each stakeholder group that is being referred to.

• Generally, it is not clear why the results were analysed and presented separately for each stakeholder group? Why was that important and what extra value does it bring to the narrative on COVID-19 vaccine roll out in Lagos. This needs to come out clearly in the introduction and discussion especially as some on the themes are similar across stakeholder groups. If there are no specific reasons, the authors should consider reporting the results by thematic area only and supporting quotes should indicate if it was a HCW, key informant or vaccine recipient.

Reviewer #3:

This is a very interesting and important article, outlining in great detail the complexities and successes of delivering COVID-19 vaccination in Lagos. I think the article could state its contribution even more strongly, by noting its originality in bringing together perspectives of vaccine recipients and those involved at different levels of the vaccine delivery system. Emphasising the contribution will also require the authors engaging a bit more thoroughly with the social science literature on vaccine confidence and delivery - the Vaccine Confidence Project (https://www.vaccineconfidence.org/) can be a good start, and literature by Heidi Larson and colleagues includes some good overviews of the literature.

The area where I think some major revisions are required is in the interpretation of the data. This will not require a great deal of re-analysis, but rather really considering what the data is saying and what it is not. It seemed to me that having interviewed vaccine recipients and providers, the authors cannot in fact suggest to have insight into drivers of vaccine hesitancy. Discussions of conspiracies and mistrust are *reported* by respondents but not in fact stated as reasons for *their* refusal. They are in other words speculations about low uptake. For example on page 20 you state “the poor turnout for vaccination was because awareness was low”— but you haven’t actually explored reasons for refusal/hesitancy with those who refused, so you can only really say that poor turnout was perceived by key informants to be linked to low awareness.

Research has shown that healthcare providers might overstate low awareness when reasons for hesitancy are more complex (see for example Enria et al "Bringing the social into vaccination research" in PLOS One). It does not mean that the fact that providers and vaccine recipients deem this to be significant is not worth stating, but it needs to be interpreted as their speculation rather than as a finding that this is what you take to be the most important issue affecting vaccination uptake. The respondents' description of their encounters of misconceptions in practice are valuable, but need to be interpreted as such, given that they did not in fact drive their own vaccination behaviour (everyone you interviewed is pro vaccination). Is there any data available on uptake in the facilities you studied and the relative significance of supply and demand issues? On page 27 you note “poor uptake”—can this be quantified? Were any access issues reported in your findings? Is there any research you could cite that offers more concrete evidence on the hesitancy problem, if this is indeed a large problem in practice? I thought that it was extremely interesting that your respondents worried there might not be enough vaccines for everyone—how does this relate to concerns about mistrust?

Related to the question of potential hesitancy, I would suggest that you consider a more nuanced interpretation of mistrust in vaccination. There may be more complex reasons why people have low trust in vaccination, and indeed there is very good literature on this from Nigeria as well, which situates for example the polio controversy (see Obadare E. A crisis of trust: history, politics, religion and the polio controversy in Northern Nigeria. Patterns Prejudice. 2005;39(3):265–84.) or broader literature that contestes the “knowledge deficit” model (see Goldenberg M. Public Misunderstanding of Science?: Reframing the Problem of Vaccine Hesitancy. Perspect Sci. 2016;24(5):552–81.). You can consider this in the discussion, but as noted above, I would also treat the issue of mistrust much more carefully, with a more nuanced interpretation and perhaps not make it your central finding as I am not sure that is what your data suggests.

Some smaller isses that you could fix:

- Because you are reporting data from very different kinds of respondents, I would make sure that throughout you make clear which you are speaking of—so for example on page 20 instead of saying “most of the respondents”, clarify that this is among key informants

- P.5 why is this study framed as a “risk assessment”?

- How were the 8 sites selected?

- Clarify the categories (i.e. HCWs were also “vaccine recipients” but I assume you are differentiating? Are there any more demographic characteristics on vaccine recipients? How were they selected?)

- You mention using a theoretical framework—what was the framework used?

- I would make clear from the beginning that vaccine recipients were interviewed through FGDs, HCWs and Key informants through interviews. Maybe you could move Table 1 to the methodology so it’s clearer from the start.

Reviewers' comments:

Reviewer's Responses to Questions

**Comments to the Author**

1. Does this manuscript meet PLOS Global Public Health’s publication criteria? Is the manuscript technically sound, and do the data support the conclusions? The manuscript must describe methodologically and ethically rigorous research with conclusions that are appropriately drawn based on the data presented.

Reviewer #1: Yes

Reviewer #2: Yes

Reviewer #3: Yes

2. Has the statistical analysis been performed appropriately and rigorously?

Reviewer #1: N/A

Reviewer #2: N/A

Reviewer #3: Yes

3. Have the authors made all data underlying the findings in their manuscript fully available (please refer to the Data Availability Statement at the start of the manuscript PDF file)?

Reviewer #1: Yes

Reviewer #2: No

Reviewer #3: No

4. Is the manuscript presented in an intelligible fashion and written in standard English?

Reviewer #1: Yes

Reviewer #2: Yes

Reviewer #3: Yes

5. Review Comments to the Author

Reviewer #1: Overall

This paper focuses on the COVID-19 vaccination rollout in Lagos from the perspectives of vaccine recipients, HCWs, and government officers. Overall, the authors do a great job providing and presenting a lot of data on the rollout; I particularly like the inclusion of different perspectives. Some of the language can be cleaned up throughout, but overall, I think it’s a paper the authors should be proud of and is one that will benefit future vaccine rollouts in Nigeria and elsewhere.

Some detailed comments are provided:

Introduction

- Line 80: Provide a little more detail – perhaps the number of vaccines currently produced (can say “at the time of writing”) to provide context of scale

- Lines 80-84: Can you be more specific? Is this clinical data? Consider breaking the sentence into two separate ones for clarity.

- Line 84: were the blood clots discredited, or was it reports that it was unsafe? Please clarify.

- Line 85: Is it possible to be more specific than “Africa”? Maybe “Vaccination against COVID-19 has been slower in countries relying on doses from COVAX than in other places” or something like that.

- Lines 88-89: This is a very important point, but the sentence is unclear. What proportions are you speaking of?

- Line 90: An example of an exemplary rollout would be helpful.

- Lines 91-95: This sentence needs some citation(s), as there have been several reports about why the rollouts/administration may have been delayed – including, but not listed here, countries receiving vaccines right before they were to expire.

- Line 96: Which vaccine was provided?

- Line 106: Did Lagos State receive their share immediately?

- Line 124: When did the vaccine become available to all adults?

- COME BACK TO AIM/LINE 125

Methods

- Overall, the divisions and sub-divisions are a bit unclear. Consider combining some sections or making note of how sub-sections may fit into broader sections (using numbers and/or letters).

- Line 134: Lagos city or Lagos state?

- TOO MUCH INFO? Perhaps link to rollout more and cut the rest… And then connect with study sites

- Lines 158-162: What does it mean to be formally approached? How was that different than approaching vaccine recipients?

- Data collection tools and techniques: I’m a little confused by “qualitative inquiry.” You mean the qualitative data were collected using a guide, correct? Was this guide structured or semi-structured? Please provide a little more detail on the key topics addressed in the guides and how they differed between IDIs and KIIs. Same can be said for the FGDs – please provide a little information on key topics covered.

o Where FGDs held the same day as the recruitment of participants? Or did participants return on a separate day?

- Line 180: The number of IDIs, etc. carried out should be included in the results section. Also, if there were 8 sites and 4 IDIs were carried out at each site, why were 24 conducted (rather than 32)? 24 is still a lot of IDIs, but please comment if there were decliners or a reason why the numbers do not match.

- Lines 183-186: Is there a way to not include their titles? I imagine this makes the KII participants easy to identify… Please be more general in how they are referred to.

- Lines 191-192: Please clarify. “Transcribed verbatim” means word-for-word, so by the rest of the sentence, do you mean that the other notes (e.g. non-verbal cues, etc.) were included into the transcripts as well?

- Line 196: “manuscripts” should be changed to “transcripts”

- Lines 199-207: There is a lot of detail here that I believe can be written/edited in a way that is more succinct and clearer to the reader.

o If no coding software was used, did the authors use Word or Excel?

Results

- The number of KIIs, IDIs, FGDs should be included at the beginning of the results section. Here you can also include some summarized information on the participants (gender, age, etc.)

o Were men and women included in the same FGDs?

o What is the gender breakdown of the IDIs and KIIs?

o I’m still worried that by providing their titles, the identity of the KIIs is easy to figure out.

- Lines 252-254: did the participants themselves have concerns about the government’s motives or the uncertainty of the effect, or was it what they had heard from others? If they had these concerns, what led them to still get vaccinated?

- In this section (myths/misconceptions), it would be helpful to have a specific example or two. Participants seem to have heard of the myths (did they mention any specific myths?) but still got vaccinated. Did anyone say why that was?

o OK, this is mentioned in the next section. Was it the same respondents?

- Liens 303-305: Was this hesitancy the HCWs had seen while working in the clinic – was it directly observed by them? Or was this more of an explanation for wider vaccine hesitancy?

- Lines 308-311: Were participants asked to score the programme during the IDIs? Was this a formal assessment, or was it part of the interview questions? If it was formal, please say more on that and include it in the methods section.

- Line 323: should there be “not” in that sentence?

- Lines 339-351: Since the steps are already mentioned in the intro, I think this section can be cut. Or, the focus should not be on repeating the steps, but on the KII participants’ views on how this went, barriers/successes, etc.

- Section from KIIs in general, lines 331-461: I have a few concerns about this section.

o First, there is a lot of space given to what was only 3 KIIs in comparison to the IDIs and FGDs. Much of the first part is just a summary of the process that occurred, much of which is included in the intro. I think what is most relevant for the study can stay in the intro and then the rest can be removed and instead placed in a figure or text box, possibly to be included in the annex or as supplementary information.

o The parts that bring up the KII participants’ opinions on what went well and what can be improved upon are good and could be elaborated on – especially as their comments relate and compare to what the IDI participants said.

o Lines 423-425: is this the opinion of the participants? As it reads, it sounds almost like they are summarizing a study they did.

o I’m not sure what is different between “barriers” and “challenges.”

- In general, the sub-sections of the Results all follow the same order of plan, process, and progress for each group of participants separately. Something to consider would be blending the results from the different participants so they can be compared and contrasted as it relates to this overall main outline (plan, process, progress). This would keep make the results more succinct, allow for comparison between different groups of respondents, and make the sections and sub-sections easier to follow.

Discussion

- Overall, there is a lot of information in the discussion, but I feel it’s typically very broad and general. I think this section would benefit a lot from clear examples in the literature and how those examples link directly with the data presented in the results.

- Lines 480-481: Sentence needs to be written more clearly.

- Lines 502-507: Are there other reasons why certain sites had better turnout than others? Was it only do to misinformation, or are there structural factors that may play a role?

- Lines 507-508: Citation?

- Lines 513-516: Can you be more specific? Perhaps include more of the information participants gave regarding how or what they want included in vaccine messages and how the rollout may address those issues in the future.

- Lines 517-520: I’m not sure I understand this sentence… How does high morbidity/mortality lead to a reduction in vaccine hesitancy? Also, this is a rather old source – is there anything newer that addresses this issue, particularly as it may relate to COVID?

- Lines 526-531: Is it the AEFIs themselves, or people’s perceptions of the AEFIs that is most important for vaccine uptake?

- Lines 536-541: I find that participants wanted a more bottom-up approach interesting. Was this something that the KII participants discussed too? Is there a way to compare/contrast this desire with the plans of the rollout?

- Lines 542-548: Have there been any studies about incorporating other vaccines into routine immunization?

- Lines 552-556: Can you provide an example of the use of these types of partnerships? How would COVID vaccine rollout be similar or different from these examples? Are there importance lessons that can be applied?

- Overall, how can other countries learn from Nigeria? This study gives details on many challenges and lessons learnt – how can other countries benefit from this information? How can Nigeria benefit from this information in the future?

Reviewer #2: Re: A qualitative analysis of the COVID-19 vaccination rollout in Lagos, Nigeria: Client and

provider perspectives on the plan, the process and the progress

This paper reports on a qualitative study that assessed the plan, process, and progress of the COVID-19 vaccination program in Lagos State Nigeria. Using in-depth interviews and focus group discussions, the authors highlight, from the perspectives of vaccine recipients and healthcare workers, the challenges to the COVID-19 vaccine implementation program in Lagos State including issues around vaccine hesitancy and logistics. Factors that contributed to the success of the program are also presented. The study is novel and has interesting and important findings that could be used to inform future vaccine programs in Lagos and Nigeria generally

The presentation of the paper however needs to be improved in terms of structure, additional information in the Method’s section, and support (quotes from IDIs and FGDs) for some of the key results. My suggestions are as follows:

• Going through the results and certain sections of the abstract, it became less and less clear if the manuscript is reporting on the outcome of research or a monitoring and evaluation exercise. Perhaps this is because of the way the results are presented (especially the choice of words). For example, participants agreed, commending the government of Nigeria, and detailed descriptions of the actual roll out program instead of a presenting the perspectives of the different stakeholder groups without the authors qualifying them as: “genuine” “successful, “unfounded claims” etc. Also, where the authors present the results of IDIs with healthcare workers and key informants, there is an inclination to provide a defence to most of the challenges mentioned by the interviewees. That may be okay, if those reasons are provided in the discussion section for example… although in the interviews, participants mentioned that XXX was a major logistic-related challenge, from our knowledge and experience of the roll out program, this challenge may have been due to XX and was resolved by XX.

• Following up from the point above, it is important for the authors to describe their positionality in the context of this research. For example, where some of them involved in the vaccine roll out program and if so in what capacity and how this may have influenced the results or discussions. This will give researchers some perspectives on the overall narrative and arguments advanced in the paper.

• The methods section needs more information:

• Line 159: vaccine rollout process or vaccine roll out program?

• Line 160: It seems to me like two sampling strategies were used. Purposive (frontline workers) and convenience (clients/vaccine recipients).

• 161: Participants were formally approached- What does this mean? -A description of the process and why that was necessary will provide more clarity

• 164: Delete “For the qualitative inquiry”. The title already states that it is a qualitative study. Also, only qualitative methods were used

• 164: Moderator’s guide: What was the nature of this guide? Semi structured open ended? close ended? What type of questions were included and how was it developed?

• 165: State number of IDIs and KII

• What was the difference between the two? was it in the method or in the stakeholder group interviewed? If methods, then I suggest that should be described. If it is in the stakeholder group, then I suggest you use IDIs across the paper

• The methods section should have a justification of why different methods were used for the different stakeholder groups. For example, why FGDs with vaccine recipients and not IDIs. Similarly, why IDIs with frontline workers and not FGDs. Also, it needs to be stated clearly at the introduction that FGD were conducted with vaccine recipients only and IDIs with healthcare workers and key informants (Administrators?) only. Otherwise, it becomes confusing who was involved in what-especially in the results section.

• How long were the IDIs and KIIs? This information is provided for FGDs but not the other methods.

• How did the authors arrive at the final sample size for each stakeholder group and in the case of FGDs, why 8 participants per FGDs?

• What was the overall study period? When were the FGDs conducted, was it immediately after vaccination? If yes, how were participants selected from the overall pool of persons who presented at the vaccine site on that specific day?

• What was the criteria for selecting the key officers? That is how did the team decide who was a “key informant”? for example why the state epidemiologist and not the Commissioner of Health or the Head of the Nigeria CDC?

• Line 188: Were all the Investigator part of the training or only those responsible for the conducting the FGDs, KIIs etc If not all, consider indicating the initials of those who were trained for this purpose.

• Line 195-196: What does transferability for cultural reasons mean and what impact did this have on the thematic coding and data analysis.

• Line 196: Read the manuscript (transcript?)

• Line 199: A description of the theoretical framework is important to give readers a perspective of the analysis. Is it an existing framework that has been published or used in similar studies or was it developed during the coding process by those involved in the analysis? - If so, was it developed etc. It will also be important to state if the thematic coding was inductive or deductive.

• Line 200: The sentence, “Next, by the process of indexing. This process of indexing needs to be described or referenced for the benefit of your readers.

• Line 203: what is understandable data? Either provide a description or provide a reference of what it means in the context of qualitative research.

• Line 205: Were the several meetings with the four team members involved in the coding or all the authors.

• Line 213: Purely voluntary- I suggest purely be deleted except it goes beyond voluntary.

Results:

• Table 1: Headings in Column 1 are not consistent-In one instance the method is used, in another the name of the stakeholder group. I suggest the authors use 1 across all groups, preferably the stakeholder group (Vaccine Recipients; Front line health care workers etc)

• Overall, the presentation of the results is confusing. It is presented according to the Methods. I suggest the headings should rather be defined by the different Stakeholder Groups. For example, “Perspectives of vaccine recipients on the COVID-19 vaccine rollout Program in Lagos”; “Perspectives of frontline healthcare workers on the COVID-19 vaccine rollout Program in Lagos” etc.

• Line 234: Introduce a quote from the FGDs to support the statement.

• Line 245: The theme needs to be rephrased to capture the goal of the project (COVID-19 vaccine roll in Lagos). For example, “Myths or misconceptions about COVID_19 Vaccine and the impact on the vaccine-roll out program in Lagos State”. The paper is on the vaccine roll out process and not COVId-19 vaccine hesitancy. How it leads to hesitancy can then be capture in the narrative rather than the sub-heading (a key theme)

• Line 249: A supporting quote should be inserted at the end of the sentence “…… conspiracy theories about theCOVID-19 vaccine” to support the analysis/narrative

• Line 249: Sentence starting with Unfounded claims…. I suggest that as much as possible, the authors should remain value neutral in presenting the results and avoid the use of words like “unfounded” except that was used by the participants and there is a supporting quote for that. That phrasing, if necessary, can be used in the discussion section with a supporting argument on why they are considered “unfounded”. I think here is an opportunity by the authors to educate, rather than then dismiss a claim as unfounded” because such dismissals could entrench covid-19 vaccine hesitancy or resistance.

• Line 250: Participants mentioned that..….. Was this all or some, or most of the participants? This applies across the manuscript.

• Line 267. Sentence ending with “achieved faster” needs a supporting quote. Throughout the results, please provide supporting quotes for each result.

• Line The statement “It was also opined that vaccination would help improves the economy…” also requires a supporting quote

• A mentioned earlier, consider replacing In-depth Interviews with a description around the lines “Frontline Workers' Perspectives on COVID-19 Vaccines the roll out program in Lagos Nigeria”

• Why are supporting quotes from the IDIs not reported in the same way as the FGDs? The FGDS for example has the gender and age of the participants in addition to the unique participant number. While the IDIs only have the unique IDs. This needs to be consistent across the paper. Analysis in terms of age and gender are also not presented, so it is hard to tell why the authors reported the FGDs using gender and age.

• Lines 314-318: Up to three quotes are provided to support the results that HCWs commended the government, and all quotes are very similar. I suggest one quote is used for this purpose especially as in most cases, just one quote is provided for a key result- and in other cases no quotes.

• Lines 320- Kindly provide description on the specific aspect(s) of planning that came up from the interviews before providing a quote. One quote should be okay.

• Line 331: Instead of key informant interviews, consider using “Providers Perspectives on COVID-19 Vaccine Roll out in Lagos Nigeria”

• Line 332: Considering the length of this manuscript and readers may have forgotten which stakeholder groups were described as “Providers”, It is important, for the purpose of clarity, to repeat that information in this sentence - May be important to repeat it here for clarity.

• Line 339-351: I suggest the paragraph be deleted as it doesn’t form part of the key results nor adds value to the results. Same for like 357-359.

• Line 367: Please provide a supporting quote.

• Line 370-71: Why was there a need to review the process? I think that is the key information as the article reports on vaccine roll out process.

• Line 386: Issues around logistics has already been reported. Also, Lines 397-407 seems to be an evaluation of the logistics rather than the perspectives of the key informants. I therefore suggest that the paragraph be deleted.

• Lines 442: Please provide supporting quote

• Lines 443: For the entire section, please provide supporting quotes for each of the key results.

• Lines 450-453: It is not clear how failure to observe COVID-19 protocols by the already vaccinated, was a challenge to the vaccine roll out program in Lagos. That will need further explanation.

• Lines 469: the word client has only been introduced in the discussion. My guess is that this refers to vaccine recipients. However, I think, the authors should opt for one description for that stakeholder group and use it throughout the manuscript.

• Overall, the use of “participants” across the results section is confusing, I suggest it should be replaced with the title of each stakeholder group that is being referred to.

• Generally, it is not clear why the results were analysed and presented separately for each stakeholder group? Why was that important and what extra value does it bring to the narrative on COVID-19 vaccine roll out in Lagos. This needs to come out clearly in the introduction and discussion especially as some on the themes are similar across stakeholder groups. If there are no specific reasons, the authors should consider reporting the results by thematic area only and supporting quotes should indicate if it was a HCW, key informant or vaccine recipient

Reviewer #3: This is a very interesting and important article, outlining in great detail the complexities and successes of delivering COVID-19 vaccination in Lagos. I think the article could state its contribution even more strongly, by noting its originality in bringing together perspectives of vaccine recipients and those involved at different levels of the vaccine delivery system. Emphasising the contribution will also require the authors engaging a bit more thoroughly with the social science literature on vaccine confidence and delivery-- the Vaccine Confidence Project can be a good start, and literature by Heidi Larson and colleagues includes some good overviews of the literature.

The area where I think some major revisions are required is in the interpretation of the data. This will not require a great deal of re-analysis, but rather really considering what the data is saying and what it is not. It seemed to me that having interviewed vaccine recipients and providers, the authors cannot in fact suggest to have insight into drivers of vaccine hesitancy. Discussions of conspiracies and mistrust are *reported* by respondents but not in fact stated as reasons for *their* refusal. They are in other words speculations about low uptake. For example on page 20 you state “the poor turnout for vaccination was because awareness was low”— but you haven’t actually explored reasons for refusal/hesitancy with those who refused, so you can only really say that poor turnout was perceived by key informants to be linked to low awareness.

Research has shown that healthcare providers might overstate low awareness when reasons for hesitancy are more complex (see for example Enria et al "Bringing the social into vaccination research" in PLOS One). It does not mean that the fact that providers and vaccine recipients deem this to be significant is not worth stating, but it needs to be interpreted as their speculation rather than as a finding that this is what you take to be the most important issue affecting vaccination uptake. The respondents' description of their encounters of misconceptions in practice are valuable, but need to be interpreted as such, given that they did not in fact drive their own vaccination behaviour (everyone you interviewed is pro vaccination). Is there any data available on uptake in the facilities you studied and the relative significance of supply and demand issues? On page 27 you note “poor uptake”—can this be quantified? Were any access issues reported in your findings? Is there any research you could cite that offers more concrete evidence on the hesitancy problem, if this is indeed a large problem in practice? I thought that it was extremely interesting that your respondents worried there might not be enough vaccines for everyone—how does this relate to concerns about mistrust?

Related to the question of potential hesitancy, I would suggest that you consider a more nuanced interpretation of mistrust in vaccination. There may be more complex reasons why people have low trust in vaccination, and indeed there is very good literature on this from Nigeria as well, which situates for example the polio controversy (see Obadare E. A crisis of trust: history, politics, religion and the polio controversy in Northern Nigeria. Patterns Prejudice. 2005;39(3):265–84.) or broader literature that contestes the “knwoeldge deficit” model (see Goldenberg M. Public Misunderstanding of Science?: Reframing the Problem of Vaccine Hesitancy. Perspect Sci. 2016;24(5):552–81.). You can consider this in the discussion, but as noted above, I would also treat the issue of mistrust much more carefully, with a more nuanced interpretation and perhaps not make it your central finding as I am not sure that is what your data suggests.

Some smaller isses that you could fix:

- Because you are reporting data from very different kinds of respondents, I would make sure that throughout you make clear which you are speaking of—so for example on page 20 instead of saying “most of the respondents”, clarify that this is among key informants

- P.5 why is this study framed as a “risk assessment”?

- How were the 8 sites selected?

- Clarify the categories (i.e. HCWs were also “vaccine recipients” but I assume you are differentiating? Are there any more demographic characteristics on vaccine recipients? How were they selected?)

- You mention using a theoretical framework—what was the framework used?

- I would make clear from the beginning that vaccine recipients were interviewed through FGDs, HCWs and Key informants through interviews. Maybe you could move Table 1 to the methodology so it’s clearer from the start.

6. PLOS authors have the option to publish the peer review history of their article (what does this mean?). If published, this will include your full peer review and any attached files.

**Do you want your identity to be public for this peer review?** For information about this choice, including consent withdrawal, please see our Privacy Policy.

Reviewer #1: No

Reviewer #2: **Yes: **Nchangwi Syntia Munung

Reviewer #3: No

---

## [Decision Letter · Decision Letter 1]

13 Jul 2022

PGPH-D-21-01166R1

A qualitative analysis of the COVID-19 vaccination rollout in Lagos, Nigeria: Client and provider perspectives on the plan, the process and the progress

Dear Dr. Kanma-Okafor

Thank you for submitting your manuscript to PLOS Global Public Health. After careful consideration, we feel that it has merit but does not fully meet PLOS Global Public Health’s publication criteria as it currently stands. Therefore, we invite you to submit a revised version of the manuscript that addresses the points raised during the review process.

We look forward to receiving your revised manuscript.

Kind regards,

Muhammed Olanrewaju Afolabi, MD, MPH, PhD

Academic Editor

Journal Requirements:

1. In the Funding Information you indicated that no funding was received. Please revise the Funding Information field to reflect funding received.

Please ensure that the funders and grant numbers match between the Financial Disclosure field and the Funding Information tab in your submission form. Note that the funders must be provided in the same order in both places as well.

Additional Editor Comments (if provided):

Reviewers' comments:

Reviewer's Responses to Questions

**Comments to the Author**

1. If the authors have adequately addressed your comments raised in a previous round of review and you feel that this manuscript is now acceptable for publication, you may indicate that here to bypass the “Comments to the Author” section, enter your conflict of interest statement in the “Confidential to Editor” section, and submit your "Accept" recommendation.

Reviewer #1: (No Response)

Reviewer #3: (No Response)

2. Does this manuscript meet PLOS Global Public Health’s publication criteria? Is the manuscript technically sound, and do the data support the conclusions? The manuscript must describe methodologically and ethically rigorous research with conclusions that are appropriately drawn based on the data presented.

Reviewer #1: Yes

Reviewer #3: Yes

3. Has the statistical analysis been performed appropriately and rigorously?

Reviewer #1: N/A

Reviewer #3: N/A

4. Have the authors made all data underlying the findings in their manuscript fully available (please refer to the Data Availability Statement at the start of the manuscript PDF file)?

Reviewer #1: Yes

Reviewer #3: No

5. Is the manuscript presented in an intelligible fashion and written in standard English?

Reviewer #1: Yes

Reviewer #3: Yes

6. Review Comments to the Author

Reviewer #1: Please see attachment

Reviewer #3: Please see uploaded comments

7. PLOS authors have the option to publish the peer review history of their article (what does this mean?). If published, this will include your full peer review and any attached files.

**Do you want your identity to be public for this peer review?** For information about this choice, including consent withdrawal, please see our Privacy Policy.

Reviewer #1: No

Reviewer #3: No

---

## [Editor Report · Decision Letter 2]

12 Oct 2022

A qualitative analysis of the COVID-19 vaccination rollout in Lagos, Nigeria: Client and provider perspectives on the plan, the process and the progress

PGPH-D-21-01166R2

Dear Dr Kanma-Okafor,

Apologies for the delays throughout this editorial and review process, and thank you for submitting your revised manuscript. We are pleased to inform you that your manuscript 'A qualitative analysis of the COVID-19 vaccination rollout in Lagos, Nigeria: Client and provider perspectives on the plan, the process and the progress' has been provisionally accepted for publication in PLOS Global Public Health.

Best regards,

Sabine Hermans

Academic Editor
